# Endothelial Progenitor Cells Promote Osteosarcoma Progression and Invasiveness via AKT/PI3K Signaling

**DOI:** 10.3390/cancers15061818

**Published:** 2023-03-17

**Authors:** Ofri Doppelt-Flikshtain, Amin Younis, Tal Tamari, Ofir Ginesin, Talia Shentzer-Kutiel, David Nikomarov, Gil Bar-Sela, Benjamin R. Coyac, Yehuda G. Assaraf, Hadar Zigdon-Giladi

**Affiliations:** 1Laboratory for Bone Repair, Rambam Health Care Campus, Haifa 3109601, Israel; 2The Ruth and Bruce Rappaport Faculty of Medicine, Technion-Israel Institute of Technology, Haifa 3525422, Israel; 3Department of Periodontology, School of Graduate Dentistry, Rambam Health Care Campus, Haifa 3109601, Israel; 4Thoracic Cancer Service, Rambam Health Campus, Haifa 3109601, Israel; 5Musculoskeletal Oncology Department, Rambam Health Campus, Haifa 3109601, Israel; 6Oncology Departmant, Emek Medical Center, Afula 1834111, Israel; 7The Fred Wyszkowski Cancer Research Laboratory, Department of Biology, Technion-Israel Institute of Technology, Haifa 3200003, Israel

**Keywords:** endothelial progenitor cells, osteosarcoma, metastasis

## Abstract

**Simple Summary:**

Although progress has been made in the treatment and survival of patients with a range of malignancies, the prognosis for patients harboring metastatic osteosarcoma (bone cancer) remains dismal due to the limited therapeutic options available. Endothelial progenitor cells (EPCs) mediate the angiogenic (blood vessel formation) switch in several cancers. Spatial proximity between EPCs and osteosarcoma in the bone led to the hypothesis that EPCs-osteosarcoma interactions may possibly promote osteosarcoma (OS) progression and aggressiveness. In the current paper, we demonstrate the non-physical role that EPCs play in OS migration and invasion, along with deciphering a potential underlying molecular mechanism. Our findings may pave the way toward the development of new EPCs-targeted therapies to inhibit OS metastasis and hence enhance the therapeutic efficacy of this devastating bone cancer.

**Abstract:**

Background: Osteosarcoma (OS) mortality is attributed to lung metastases. Endothelial progenitor cells (EPCs) mediate the angiogenic switch in several cancers. The spatial proximity between EPCs and OS in the bone led to the hypothesis that EPCs-osteosarcoma interactions may possibly promote OS progression and aggressiveness. Methods: A PI3K inhibitor, Bevacizumab (an anti-VEGF-A antibody), and an anti-FGF2 antibody were added to the EPCs’ conditioned medium (EPC-CM), and their impacts on OS cell (U2-OS and 143B) proliferation, migration, invasion, MMP9 expression, and AKT phosphorylation were determined. The autocrine role of VEGF-A was assessed using Bevacizumab treatment and VEGF-A silencing in OS cells. Toward this end, an orthotopic mouse OS model was established. Mouse and human tumors were immunolabeled with antibodies to the abovementioned factors. Results: EPC-CM enhanced osteosarcoma MMP9 expression, invasiveness, and migration via the PI3K/AKT pathway. The addition of Bevacizumab and an anti-FGF2 antibody to the EPC-CM diminished OS cell migration. The autocrine role of VEGF-A was assessed using Bevacizumab and VEGF-A silencing in OS cells, resulting in decreased AKT phosphorylation and, consequently, diminished invasiveness and migration. Consistently, OS xenografts in mice displayed high VEGF-A and FGF2 levels. Remarkably, lung metastasis specimens derived from OS patients exhibited marked immunolabeling of CD31, VEGF-A, and FGF2**.** Conclusions: EPCs promote OS progression not only by physically incorporating into blood vessels, but also by secreting cytokines, which act via paracrine signaling. EPCs induced in vitro MMP9 overexpression, invasion, and migration. Additional animal studies are warranted to further expand these results. These findings may pave the way toward the development of novel EPCs-targeted therapeutics aimed at blocking OS metastasis.

## 1. Introduction

Osteosarcoma (OS) is an aggressive primary bone cancer which stems from a mesenchymal origin. It occurs most frequently in teenagers and young adults, with a second incidence peak among individuals over the age of 60 [1,2]. The primary tumor can metastasize, predominantly to the lungs. In high-income countries, synchronous metastases are found in 10–30% of OS patients compared to 40% in low- and middle-income countries. However, due to limited imaging techniques, the exact number of patients with metastases at diagnosis could be even higher, and most patients are actually suspected to harbor micrometastases at diagnosis [3,4]. Patients with metastatic disease exhibit a dismal prognosis, with approximately a 5-year survival rate of 20–30%, compared with 80% of patients with localized disease [3,5]. Nowadays, neoadjuvant chemotherapy combined with limb sparing wide surgical resection and post-operative adjuvant chemotherapy is the treatment of choice for OS patients, while biological therapies are not included in treatment protocols [6]. The treatment strategies and the survival rates of OS have remained the same for decades [7].

Endothelial progenitor cells (EPCs) can be isolated from human blood or bone marrow. There are two distinct subpopulations of EPCs: early EPCs with myeloid/hematopoietic characteristics and late EPCs. Early EPCs produce colonies consisting of a spindle-like shape and appear early (4–7 days) in culture dishes. They have limited proliferative capacity, and they are positive for CD31, CD45, CD14, and VEGF receptor 2 (KDR), whereas late EPCs produce colonies that appear later (2–4 weeks after seeding). They form colonies of proliferative cells with a cobblestone-like morphology [8]. In addition, they are characterized by the expression of cell-membrane markers, including CD31, CD34, and VEGF receptor 2 (KDR), but are negative for CD133, CD45, and CD14. Recently, other surface markers of EPC were described, including CD144 and CD146. CD157, a tissue resident marker was also identified as a marker of tissue-resident vascular endothelial stem cells [8,9]. In addition, CXCR4 expression was found to induce EPCs’ migration to ischemic tissues; however, it is not usually used as a characteristic marker of EPCs [10]. Additional characteristics of EPCs are LDL uptake, lectin binding, and tube formation [11]. EPCs promote cancer progression via their incorporation into sprouting blood vessels. However, they were also found to promote tumor neovascularization by forming de novo blood vessels in a process termed vasculogenesis [12,13,14,15]. Clinically, EPCs are correlated with metastatic spread in some malignancies, such as breast and ovarian cancer [16,17,18]. In OS, as well as in other cancers, EPCs respond to a gradient of angiogenic cytokines, including VEGF, FGF, and SDF-A, which are released from ischemic zones, including the hypoxic tumor microenvironment (TME) [19]. They undergo mobilization to the tumor site and play a dual role. On the one hand, they contribute to tumor neovascularization. On the other hand, in addition to the physical contribution of EPCs to newly formed blood vessels, they secrete different growth factors in a paracrine signaling manner which promote tumor growth and metastasis [20].

A previous study in our lab examined the crosstalk between EPCs and mesenchymal stem cells (MSCs) during physiological osteogenesis [21]. The study implies that stromal-cell-derived factor-1 (SDF-1), which is secreted by endothelial cells, plays a pivotal role in bone formation by stimulating the migration of MSCs. Along this vein, we herein propose that EPCs communicate with OS cells in a paracrine manner, which may contribute to OS progression, migration, and invasion.

To the best of our knowledge, although EPCs’ contribution to tumor progression has been widely investigated in different malignancies, the impact of EPCs on OS migration and invasion has yet to be examined. Therefore, in the current study, we aimed to determine whether EPCs can promote OS invasion and migration and studied the underlying molecular mechanism. Our new findings support the development of novel EPCs-targeted therapies aimed at blocking OS-EPCs interactions, thus inhibiting tumor growth and metastasis.

## 2. Materials and Methods

### 2.1. Isolation, Expansion, and Characterization of EPCs

EPCs were isolated from the blood of healthy volunteers who signed informed consent, as previously described [22]. The experiments were performed in accordance with Rambam Health Care Campus’s Helsinki Committee for human experiments (Helsinki number 0397-12 RMB). Briefly, blood was diluted at a ratio of 1:1:1 with phosphate-buffered saline (PBS) and lymphoprep (Fresenius Kabi Norge AS, Axis-Shield PoC AS, Oslo, Norway) and centrifuged (Hermle Z 300, ©HERMLE Labortechnik GmbH, Wehingen, Germany) at 750× *g* for 30 min at room temperature. The buffy coat was collected and washed twice with PBS. Pelleted cells were then suspended in an endothelial growth medium (EGM-2, Clonetics: Lonza, Basel, Switzerland) containing 20% fetal bovine serum (FBS) and penicillin–streptomycin (Biological Industries Ltd., Beit Haemek, Israel) and supplemented with all the supplied endothelial growth medium supplements (EGM-2MV SingleQuote; Clonetics, Lonza, Basel, Switzerland). Cells were seeded on six-well plates coated with fibronectin (Biological Industries Ltd.) and cultured at 37 °C in a humidified atmosphere of 95% air/5% CO_2_. After 4 days, non-adherent cells were washed out with PBS, and fresh EGM-2 medium was added. Cells were cultured until the first colonies of late EPCs appeared, usually after 2–3 weeks. EPCs were fed three times weekly and split at ~80% confluence, using trypsin (Biological Industries Ltd.) Human EPCs were characterized using flow cytometry, as follows: 2 × 10^5^ cells were suspended in 50 µL fluorescence-activated cell sorter (FACS) buffer (PBS containing 0.5% FBS) and labeled with 0.2 mg/mL of anti-human antibodies: CD31 (clone 1D2-1A5 LifeSpanBioSciences, Seattle, WA, USA), CD34 (clone 581, BD Biosciences, San Jose, CA, USA), CD45 (clone HI30, BD Biosciences, San Jose, CA, USA), VEGFR-2 (clone #89106, R&D SYSTEMS, Minneapolis, MN, USA), and CD14 (clone M5E2, BD Biosciences, San Jose, CA, USA). OneComp eBeads (Thermo Fischer Scientific, Waltham, MA, USA) were stained with 1 µL of each different fluorochrome and then used as single-color compensation controls. Cells were analyzed using cyan flow cytometry (Beckman Coulter, Brea, CA, USA). Single cell data were analyzed using FlowJo, LLC software (BD Biosciences, San Jose, CA, USA), which gave a percentage for positive cells (see Appendix A, “Characterization of Endothelial Progenitor Cells”) [21,22,23].

### 2.2. Osteosarcoma Cell Line Cultures

Human OS cell lines U2-OS and 143B (ATCC, Gaithersburg, MD, USA) were cultured according to the manufacturer’s instructions. U2-OS cells were cultured in low glucose Dulbecco’s Modified Eagle Medium (Low DMEM) supplemented with 10% FBS, 1% penicillin–streptomycin–amphotericin B, and 1% glutamine (Biological Industries Ltd.). 143B cells were cultured in Minimum Essential Medium Eagle (MEM-Eagle) with 0.015 mg/mL 5-bromo-2′deoxyuridine (SIGMA-ALDRICH, Burlington, MA, USA) supplemented with 10% FBS, 1% penicillin–streptomycin–amphotericin B, and 1% glutamine (Biological Industries Ltd.). Cells were cultured at 37 °C in a humidified atmosphere of 95% air/5% CO_2_. Cells were fed twice a week, and when reachinhg 80–90% confluence, they were split by trypsinization, using a solution containing 0.5% trypsin in 0.25% ethylenediaminetetraacetic acid (EDTA) (Biological Industries Ltd.).

### 2.3. Preparation of EPC Conditioned Medium (CM)

First, 1 × 10^6^ human EPCs were cultured in standard medium (EGM-2) until 80% confluence. Following an incubation for 48 h, 10 mL of EPC conditioned medium (EPC-CM) was collected and centrifuged to remove cells (250× *g*, 5 min) and then concentrated using a centrifugal filter (Merck Millipore, Tullagreen, Ireland) for 10 min in 2500× *g*.

### 2.4. Invasion Assay

First, 4 × 10^4^ U2-OS cells were seeded on top of the porous membrane that was coated with 100 μL Matrigel (Matrigel^®^ Corning, Bedford, MA, USA). Cells were cultured in 200 μL starvation Low DMEM medium (0.5% FBS, 1% penicillin–streptomycin–amphotericin B, and 1% glutamine) (Biological Industries Ltd.) for 4 h. Then the lower chamber was filled with Low DMEM/EGM-2 or EPC-CM. EGM-2 is a very rich medium; therefore, it was selected as a control for the experiment. OS growth medium was chosen as an additional control. Following an overnight incubation, the membrane between the two compartments was fixed with a 4% formaldehyde solution, buffered at pH 6.9 (Bio-Lab, 0006450323F1, Jerusalem, Israel), and stained with a crystal violet solution (SIGMA-ALDRICH, Burlington, MA, USA); cotton swabs were then used to remove non-migratory cells. The number of invasive cells was determined under a microscope (ECLIPSE Ts2R, Nikon, Melville, NY, USA).

### 2.5. Scratch Wound Healing Assay

First, 4 × 10^4^ U2-OS or 7 × 10^4^ 143B cells were seeded in graduated 96-well plates from ESSEN (Biological Industries Ltd.). The scratch-wound healing assay examines both proliferation and migration. However, the experiments that were conducted were performed for 16 h, which is less than the cells’ doubling time. In addition, the cells were seeded in starvation medium (0.5% FBS). When the cells reached 95% confluence, a wound was introduced on every well, using Wound Maker 96 (Essen BioScience, West Morgan Road, Ann Arbor, MI, USA), and medium was replaced by the following groups of media: Low DMEM/MEM-Eagle/EGM-2/EPC-CM. Cell migration into the wounds was monitored every 2 h and analyzed by the ESSEN IncuCyte system (Essen BioScience, MI, USA).

### 2.6. VEGF-A and FGF2 Neutralization and PI3K Pathway Inhibition

Neutralization of VEGF-A and FGF2 in EPC-CM was achieved by Bevacizumab (Avastin^®^) and an anti FGF2 antibody (AF-233-SP), respectively. For the scratch-wound healing assay, human OS cells (U2-OS and 143B) were cultured for 16 h in EPC-CM or standard OS growth medium supplemented with Bevacizumab (2 mg/mL), anti-FGF2 (0.08 µg/mL), or their combination, immediately after introducing the scratch. Furthermore, phosphoinositide 3-kinase (PI3K) inhibitor (LY-294002, Sigma-Aldrich, Burlington, MA, USA) was added at two concentrations, namely 10 and 50 µM, to the EPC-CM [24]. For the invasion assay, U2-OS cells were cultured in 200 μL starvation medium supplemented with the PI3K inhibitor (10 and 50 µM). Scratch and invasion assays were performed as described above (see Section 2.4 and Section 2.5).

### 2.7. RNA Isolation and Quantitative Real-Time PCR

OS cells (U2-OS and 143B) were cultured in EPC-CM, EGM-2, or standard OS growth medium (Low DMEM or MEM-Eagle). Total RNA was extracted from cell pellets with the HP Total RNA kit (Omega BIO-TEK, Norcross, GA, USA). A total of 1 µg RNA was used for the RT reaction. The cDNA was generated using a High-Capacity cDNA Reverse Transcription Kit (Quanta-bio, Beverly, MA, USA). Quantitative real-time PCR analysis (qPCR) was performed using SYBR green (Fast SYBR™ Green Master Mix, Applied Biosystems™, Foster City, CA, USA) for matrix metalloproteinases-9 (MMP9). Specific primers for MMP9 and housekeeping genes were used (see Appendix A, “Real-Time PCR Primer Sequence”). The results were analyzed using a Profiler PCR Array data analysis tool and normalized to generate fold change for each gene, using the ∆∆Ct method [21].

### 2.8. Western Blot Analysis

U2-OS and 143B cells were incubated for 1 h with EPC-CM, EGM-2, or standard OS medium (U2-OS: Low DMEM, 143B: MEM-Eagle). Total protein was extracted from cells by using RIPA lysis buffer containing protease (11697498001, ROCHE, Rehovot, Israel) and phosphatase inhibitor (SC-45065, Santa Cruz Biotechnology, Dallas, TX, USA). Protein samples were maintained on ice for 30 min and subsequently centrifuged (16,000× *g* for 15 min at 4 °C). Total protein was quantified using a Bradford dye regent (Cat. #5000205, BIO-RAD, Hercules, CA, USA). The extracted proteins were mixed with loading buffer and boiled at 95 °C for 5 min for protein denaturation. Protein aliquots (70 µg) were resolved using Mini-PROTEAN TGX 4–20% gradient polyacrylamide gel electrophoresis (Cat #4561094, BIO-RAD, Hercules, CA, USA). The separated proteins were subsequently transferred onto a cellulose nitrate membrane (Cat. No 10401383, Tamar, Mevaseret-Zion, Israel) and blocked for 1 h with 5% BSA (Sigma Aldrich, Rehovot, Israel) at room temperature. The membranes were incubated with the following primary antibodies overnight at 4 °C: anti-p-AKT (1:500), anti-AKT (1:1000), and anti-GAPDH (1:1000). All of these primary antibodies were diluted in 5% BSA. Membranes were washed three times with TTBS buffer (cat No. 002089232300, Bio lab, Jerusalem, Israel). Following the primary antibody incubation, membranes were incubated with anti-rabbit antibodies (146352, 147170, Cell Signaling Technology, Inc., Danvers, MA, USA), which were diluted in TTBS solution for 1 h at room temperature. Protein bands were visualized using the BioSpectrum Imaging system, and ImageJ software was used for scanning densitometry.

### 2.9. Enzyme-Linked Immunosorbent Assay (ELISA)

VEGF-A levels were determined in U2-OS and 143B cell supernatants and were compared to OS cell growth medium, using the enzyme-linked immunosorbent assay (ELISA). Following an incubation of 48 h, 10 mL of OS cells supernatant and OS standard growth medium (Low DMEM and MEM-Eagle) were collected and centrifuged to remove cells (250× *g*, 5 min) and then concentrated using a centrifugal filter (Merck Millipore, Tullagreen, Ireland) for 10 min at 2500× *g*. Quantikine^®^ ELISA kit (R&D Systems, McKinley Place, MN, USA) was used according to the manufacturer’s protocol, and the optical density was measured by a spectrophotometer at 450 nm.

### 2.10. Silencing VEGF Gene Expression

VEGF-A gene silencing was achieved using GIPZ Lentiviral Human VEGF-A shRNA kit (Horizon, Huddersfield, UK). Then, 1.5 × 10^5^ U2-OS or 143B cells were seeded in 6-well plates. After 24 h, cells were treated with starvation medium supplemented with 4 µL of viral particles for another 6 h. Then, 1 mL of complete OS growth medium (Low DMEM for U2-OS or MEM-Eagle for 143B cells) was added to each well. Following 48 h of incubation, puromycin was added at a final concentration of 5 µg/mL for 6 days to achieve selection of transduced cells. GIPZ Non-Silencing Lentiviral shRNA was used as a negative control, and cells’ transduction was conducted as mentioned above. Transduction efficiency was evaluated using real-time PCR, as described above.

### 2.11. Proteomics Analysis of EPCs Conditioned Medium

First, 10^6^ EPCs obtained from 4 independent healthy donors were cultured in serum-free medium (EGM-2) for 48 h. Conditioned medium was collected for proteomics analysis. Aliquots of 100 µL of the sample were added to 8M urea and reduced with 2.8 mM dithiothreitol (DTT) (60 °C for 30 min); supplemented with 8.8 mM iodoacetamide in 100 mM ammonium bicarbonate (in the dark, room temperature for 30 min); and digested overnight at 37 °C in 2M urea, 25 mM ammonium bicarbonate with modified trypsin (Promega) at a 1:50 enzyme-to-substrate ratio. An additional second trypsinization was performed for 4 h. The tryptic peptides were desalted using C18 tips, dried and resuspended in 0.1% formic acid. The peptides were resolved by reverse-phase chromatography on 0.075 × 180 mm fused silica capillaries (J&W) packed with Reprosil reversed-phase material (Dr Maisch GmbH, Germany). The peptides were eluted with linear 60 min gradient of 5 to 28%, 15 min gradient of 28% to 95%, and for 15 min at 95% acetonitrile with 0.1% formic acid in water at flow rates of 0.15 μL/min. Mass spectrometry was performed by Q Exactive plus mass spectrometer (ThermoFischer Scientific, Waltham, MA, USA) in a positive mode, repetitively using full MS scan, followed by collision, to induce dissociation (HCD) of the 10 most dominant ions selected from the first MS scan. Mass spectrometry data were analyzed using Proteome Discoverer 1.4 software with Sequest (ThermoFischer Scientific, Waltham, MA, USA) and Mascot (Matrix Science) algorithms against human proteome from the Uniprot database with 1% FDR. Semi-quantitation was performed by calculating the peak area of each peptide based on its extracted ion currents (XICs), and the area of the protein was the average of the three most intense peptides from each protein. PANTHER.db was used for Gene Ontology (GO).

### 2.12. Animal Model-Orthotopic Cell Injection to the Distal Femur

First, 5 × 10^5^ U2-OS cells were injected into the femur of Hsd: a-thymic nude-Foxn1nu mice. Eleven 6-weeks-old female mice (22–24 gr; (ENVIGO, Ness Ziona, Israel) were used. Cell injection and measurements of tumor size were performed by a single surgeon. Following anesthesia with 1.5% isoflurane (Piramal critical care, PA, USA) and 100% O_2_, a 27G needle was inserted into the intercondylar area of the distal femur, without incision. Cells were suspended in 20 μL PBS and injected with a 29G syringe (BD Medical, Le Pont-de-Claix, France) into the intramedullary cavity in the shaft of the femur. This 29G needle was used to inject the cells into the hole that was generated by the 27G needle. Mice were maintained under pathogen-free conditions, and buprenorphine (0.3 mg/kg bw) was injected subcutaneously 3 days post-operation. Mice behavior and mobility were monitored twice weekly. Tumor size was determined weekly by measuring the lower-limb circumference at the area of cell injection with a measuring tape. Animal weight was also determined weekly, using a digital scale. Animals were excluded from the study if body weight loss was >15%, animals appeared immobile and with closed eyes, or animals reacted violently to stimulus. Additionally, animals were sacrificed before the end of the study if tumor size was >1500 mm^3^. Following 5 and 7 weeks after OS cell injection, mice were anesthetized with isoflurane and sacrificed by cervical dislocation (N = 6 and N = 5, respectively). Tumor weight was determined by subtracting the weight of a healthy limb (contralateral) from the weight of the tumor-containing limb. Lower limbs and lungs were collected from 5 mice in the 5-week time point for histological analysis (see Appendix A).

### 2.13. Histological Preparation

Histological human samples from the primary tumors (N = 7) of non-metastatic OS patients and the metastatic tumors (N = 18) from OS patients with lung metastases were obtained from the Department of Pathology at the Rambam Health Care Campus (Ethics number 045517-RMB). Primary tumors from mice were derived as described in the Section 2.12. All samples were diagnosed and confirmed by a senior pathologist (Y.B.A). Samples were fixed with 4% paraformaldehyde. According to the sample preparation protocol of the department of Pathology at the Rambam Health Care Campus for human samples, for any tumor that involves bone, soft tumor sections are isolated and studied as non-decalcified tumors to allow molecular analyses. Therefore, none of the human samples that were used in this study (tumors from bone and lung samples) underwent a decalcification procedure. For mice samples: bone tissue samples were decalcified with gentle EDTA treatment, using Mol-DECALCIFIER (EUH210, Kalamazoo, MI, USA). Hence, both primary tumors and nearby bone (control) underwent decalcification. Paraffin blocks were sectioned (5 μm) and stained with Hematoxylin and Eosin (H&E). 

### 2.14. Immunohistochemistry

Antigen retrieval was performed at the department of Pathology, Rambam Health Care Campus, using BenchMark Ultra (Roche, Basel, Switzerland). Pretreatment conditions were set automatically for each antibody. Histological sections from human specimens were blocked with Block Buster (Background Buster, Innovex bioscience, Richmond, CA, USA) for 30 min, rinsed twice with PBS for 5 min, and immunolabeled with the following primary antibodies: anti-CD31 antibody (1:70, primary: N = 7, Metastasis: N = 18, MA5-13188, Thermo Fisher Scientific, Waltham, MA, USA), anti VEGF-A (1:200, primary: N = 7, Metastasis: N = 8, ABS82, MERCK, Millipore, Tullagreen, Ireland), and anti FGF2 antibodies (1:20, primary: N = 6, Metastasis: N = 14, NB600-1536-0.025 mL, Novus biologicals, Englewood, CO, USA) for 1 h at room temperature (see Appendix A). For the in vivo experiment, Paraffin sections from mice limbs were obtained for immunolabeling with anti-VEGF-A (1:400, ABS82, MERCK, Millipore, Tullagreen, Ireland) and anti-FGF2 (1:100, NB600-1536-0.025 mL, Novus biologicals, Englewood, CO, USA) antibodies (N = 5, Appendix A). Then the slides (human and mice) were immunolabeled for horse radish peroxidase (HRP) (ZytoChem Plus HRP Polymer anti-Rabbit/mouse, zytomed, 22 Berlin, Germany) for 30 min, rinsed, and labeled with 2,4-diaminobenzidine (DAB) (SuperPicture™ Polymer Detection Kit, DAB, rabbit, Thermo Fisher Scientific, Waltham, MA, USA) for 15 min. Following an additional rinsing, the slides were stained with Hematoxylin (10% Hematoxylin, 90% distilled water) for 1 min and washed with distilled water. To prove the specificity of the immunoreactions, samples were labeled solely with the secondary antibody, omitting the primary antibody, hence serving as a negative control (Appendix A). Blood vessels were identified, and 4–7 microscope images were taken from random microscope fields from each slide. Blood vessel density (BVD) was calculated by dividing the number of blood vessels in each image by the tumor area. Histological slides were scanned with an automatic MIDI slide scanner (3DHISTECH, Budapest, Hungary) and viewed by CaseViewer program (3DHISTECH). The quantification of VEGF-A and FGF2 labeling levels was achieved using ImageJ software.

### 2.15. Statistical Analysis

Statistical parameters, including means, medians, ranges, standard deviation (SD), and standard error (SE), were calculated. Comparisons between the groups were performed using one-way analysis of variance (ANOVA), Student’s *t*-test, and Mann Whitney U test (when samples were not normally distributed). The significance level was set at 5% and is marked with an asterisk (*). The standard errors presented in the figures are standard errors of means. Statistical analysis was performed with GraphPad Prism 9.0 (GraphPad Software, Inc., San Diego, CA, USA).

## 3. Results

### 3.1. EPCs Promote OS In Vitro Migration and Invasion via PI3K/AKT Signaling

We hypothesized that OS cells secrete factors that recruit EPCs to the tumor. EPCs induce a vascularized TME. We expected to find higher blood vessel density in aggressive OS tumors compared to low-grade non-metastatic tumors. The level of the endothelial cell marker CD31 was determined in human tumor specimens obtained from OS lung metastases and from primary OS tumors from different patients. Strikingly, blood vessel density was 30-fold higher in metastatic OS compared to non-metastatic primary tumors (*p* < 0.0001; Figure 1A,B).

We next tested the in vitro effect of EPCs on OS invasion and migration. The impact of EPC-CM on OS migration and tumor cell proliferation was assessed in a scratch-wound healing assay (U2-OS: Figure 1C–E, 143B Appendix A). Since EPC-CM failed to increase OS cell proliferation (Appendix A), we assumed that the scratch assay primarily examined the effect of EPC-CM on OS migration. Immediately after performing the scratch assay, the medium was replaced by (1) OS growth medium, i.e., control; (2) EGM-2 (endothelial progenitor medium, control); (3) EPC-CM; (4) EPC-CM + 10 µM PI3K inhibitor (PI3Ki) LY-294002; and (5) EPC-CM + 50 µM PI3Ki. EPC-CM increased U2-OS and 143B cell migration by 20–40% when compared to controls (*p* < 0.0001). PI3Ki significantly reduced OS migration in a dose-dependent manner; this PI3Ki (10 µM) decreased U2-OS migration by 10% and 50% in U2-OS and 143B cells, i.e., *p* < 0.05 and *p* < 0.0001, respectively. The higher concentration of the PI3Ki (50 µM) further decreased OS migration by 40–66% (*p* < 0.0001) (U2-OS: Figure 1C–E, 143B Appendix A). Similar findings were obtained with the Boyden chamber invasion assay: U2-OS invasion toward EPC-CM was 8.9-fold higher compared to that of both controls: U2-OS and endothelial progenitor medium (*p* < 0.0001). PI3Ki decreased the invasion capacity of U2-OS cells in a dose-dependent manner: at concentrations of 10 µM and 50 µM, the PI3Ki decreased U2-OS cell invasion by 3.5-fold and 13.6-fold, respectively, when compared to EPC-CM alone; *p* < 0.0001 (Figure 1F,G). In addition, MMP9 expression levels were increased by 1.8- and 4.5-fold in U2-OS and 143B cells, respectively, upon treatment with EPC-CM compared to EGM-2 (*p* < 0.01 and *p* < 0.05; Figure 1H and Appendix A). To further examine this signaling pathway, AKT phosphorylation levels were assessed in OS cells cultured with EPC-CM (Figure 1I,J and Appendix A). EPC-CM increased AKT phosphorylation levels by 1.6-fold compared to the EGM-2 control; however, the results attained statistical significance solely in U2-OS cells (*p* < 0.05). Collectively, these findings indicate that EPC-CM promotes OS cell migration and invasion via the PI3K/AKT signaling pathway.

### 3.2. VEGF-A and FGF2 in the EPC-CM Influence OS Migration

To pinpoint the specific EPC-CM factors that mediate MMP9 overexpression, OS migration, and invasion, we performed a mass spectrometry analysis on the EPC-CM and compared it to EPCs growth media, EGM-2 (Appendix A). This proteomics analysis revealed a *bona fide* angiogenic profile. The enriched pathways also included blood coagulation, integrin, VEGF, and FGF signaling (Figure 2A). Among the proteins that were identified in the EPC-CM were CCL-2, PDGF, SDF-1, PLGF, VEGF-A, EGF, FGF2, IGF1, and TGFβ. Thus, each protein was added alone or in different combinations to U2OS cells, and MMP9 expression was determined. The addition of VEGF-A and FGF2 enhanced MMP9 gene expression by 3.2-fold, while other factors did not alter MMP9 expression. Moreover, supplementation with 10 µM PI3Ki repressed this effect (Appendix A).

To investigate the role of FGF2 and VEGF-A in U2-OS cell migration, anti-VEGF-A antibody Bevacizumab (2 mg/mL), anti FGF2 antibody (0.08 µg/mL), and their combination were added to the EPC-CM (Figure 2). An IgG antibody was used as a negative control (Appendix A). Bevacizumab (BV) and an anti-FGF2 antibody modestly decreased U2-OS cell migration by 12% and 11%, respectively, compared to EPC-CM alone (*p* < 0.0001, *p* = 0.0063). The combination of both antibodies decreased U2-OS migration by 26%, *p* < 0.0001 (Figure 2B,C). Overall, these results show that VEGF-A and FGF2 stimulate U2-OS migration. Blocking each factor alone had a modest effect on U2-OS cell migration, suggesting that the combined blockade of both VEGFA and FGF2 is necessary to attenuate migration.

### 3.3. OS Secretes VEGF Which Promotes Its Invasiveness in an Autocrine Manner

To determine whether autocrine VEGF signaling can initiate self-sustainable cell migration and invasion, we determined VEGF-A secretion by OS cells, using ELISA. Both OS cell lines were found to secrete very high levels of VEGF-A, compared to OS growth medium: U2-OS cells by 51.1-fold, *p* < 0.01; and 143B cells by 91.8-fold, *p* < 0.0001 (Figure 3A,H). Next, we silenced VEGF-A in OS cells. Transduction efficiency was tested in real-time PCR and by analyzing GFP expression levels (Figure 3B,I). The transduction efficiency in these experiments attained >70%. VEGF-A silencing decreased U2-OS cell invasion by 10.3-fold compared to the non-targeted control (*p* < 0.0001, Figure 3C). Moreover, both VEGF-A silencing and Bevacizumab addition (added to U2-OS cell growth medium -Low DMEM) significantly decreased OS cell migration. Bevacizumab modestly decreased wound confluence by 10% (*p* < 0.05, compared to Low DMEM alone) and VEGF-A silencing by ~20% (*p* < 0.0001). The VEGF addition to shVEGF cells restored OS migration by 11–14% (U2-OS cells: *p* < 0.05 and 143B cells: *p* < 0.01) (Figure 3D,E,J,K). VEGF-A silencing also decreased AKT phosphorylation levels in OS cells by more than 2-fold compared to the non-targeted control (*p* < 0.05; Figure 3F,G,L,M).

### 3.4. High VEGF-A and FGF2 Labeling Levels in Primary Mouse OS Tumors and Human Metastatic OS Specimens

Our next step was to verify the presence of VEGF-A and FGF2 in the OS tumor before targeting these growth factors pharmacologically and genetically. We developed an orthotopic OS mouse model by injecting U2-OS cells into the femur of athymic nude mice. All mice that developed primary tumors and clinical signs of swelling were noticed after 5 weeks (Figure 4A). The average tumor perimeter was 57.0 ± 13.8 and 84.3 ± 13.9 mm after 5 and 7 weeks, respectively. Five out of six mice from the 5-week group were chosen randomly for immunohistochemistry. The immunohistochemistry analysis using anti-VEGF and anth-FGF2 showed a ~3-fold increase in VEGF-A and FGF2 labeling in primary OS tumors compared to nearby bone tissue (VEGF-A, *p* < 0.001; FGF2, *p* < 0.01; Figure 4C–F). To determine whether our findings have possible clinical significance, we determined VEGF-A and FGF2 levels in metastatic and non-metastatic OS patient biopsies. Lung metastatic OS samples exhibited significantly higher levels of VEGF-A and FGF2 compared to primary non-metastatic samples: VEGF-A by 4.5-fold, *p* < 0.05; and FGF2 by 2.5-fold, *p* < 0.0001 (Figure 4G,H). The experiment was performed using another OS cell line (143B). All the mice injected with this OS cell line induced the formation of lung metastases (Appendix A).

## 4. Discussion

Despite of the progress in the treatment and survival of patients harboring various malignancies, the prognosis of metastatic OS patients remains dismal, with limited therapeutic options [25,26]. OS arises from MSCs; therefore, investigators have invested concentrated efforts into uncovering the interactions between MSCs and OS cells. However, the bone niche is complex, and only a few studies examined OS-EPCs interactions [27,28]. Herein we show, for the first time, the non-physical role of EPCs in promoting OS migration and invasion in vitro, and we further propose a plausible molecular mechanism.

EPCs are known to undergo incorporation into the blood vessels of early stage tumors, thereby contributing to tumor progression in breast, melanoma, and lung cancer [18,29,30]. It has been shown that, following anticancer therapy, lower levels of circulating EPCs were correlated with a favorable treatment response [31]. Consistently, higher levels of EPCs before treatment were correlated with a poor prognosis [32].

Our study highlighted another important role of EPCs that is distinct from promoting tumor vessel formation. We have shown that EPCs enhance OS migration and invasion via secretion of various factors. EPC-CM did not enhance OS cell proliferation. EPC-CM is enriched with growth factors; therefore, our hypothesis was that it will promote OS cell proliferation. A possible explanation for the failure of EPC-CM to induce OS cell proliferation is the fact that we used immortalized cell lines that are often resistant to stimuli regarding their cellular proliferation. In addition, another possible explanation could be the ability of OS tumor cells to secrete growth factors in an autocrine manner. For example, in our previous study, we found that the VEGF-A concentration in the EPC-CM was 126.01  ±  22.13 pg/mL [33]. In the current study, we found very high VEGF-A levels in OS cell supernatants compared to OS growth medium. In addition to functional assays, EPC-CM was found to enhance MMP9 mRNA expression levels. In a variety of malignant tumors, MMP-2 and MMP-9 overexpression generally leads to increased tumor angiogenesis, invasion, and metastasis [34]. As for OS, MMP-9 expression was associated with an increased mortality rate [34], and the meta-analysis results suggest that MMP-9 may be a prognostic biomarker that could guide clinical therapy of OS [35,36]. MMPs also promote the secretion of proangiogenic factors by tumor cells and by the surrounding stroma [37]. In addition, activation of MMPs can be induced by several secreted growth factors, including VEGF, FGF, TGF-α and -β, and angiogenin, that can be secreted in an autocrine manner by tumor cells or in a paracrine manner from nonmalignant cells present in the TME [38].

In the current study, we also show that the secretion of VEGF-A and FGF2 promotes OS migration and invasion through the activation of PI3K and AKT, eventually leading to MMP9 overexpression (Figure 5). In this case, the concentrations of the PI3K inhibitor were very carefully selected based upon the literature; in this respect, Zhou et al. (2015) investigated the impact of the PI3K inhibitor LY-294002 on OS cells by using a very wide range of increasing PI3Ki concentrations) 5–160 µM). The results revealed that pAKT levels in cells treated with LY294002 were markedly repressed when compared with untreated cells [24]. Moreover, in an additional study (Wang et al. 2018), U2OS cells were treated with 20 µM PI3Ki; a marked inhibition of cell proliferation associated with a significant repression of the PI3K-AKT-mTOR signaling pathway was noted. Based on these studies, we used 10 µM and 50 µM LY-294002 as the working concentrations of choice [39]. Based on these studies, we used 10 µM and 50 µM as working concentrations. The Raf/MAPK and PI3K/AKT are two downstream pathways that are regulated by Ras. Both pathways are activated by growth factors (e.g., VEGF and FGF), which activate corresponding receptor tyrosine kinases (RTKs). These pathways, as well as the MYC, Notch, and the β-catenin/Wnt signaling pathways, have been identified as being frequently altered in cancer [40]. The activation of these pathways can alter the expression of key genes, and this could eventually promote tumor progression [41]. It has been previously shown that cell signaling in different malignancies is associated with an activated PI3K/AKT pathway. Activating PI3K mutations were found to be associated with increased PI3K signaling activity, resulting in enhanced tumor initiation and increased metastatic potential [42]. Moreover, enriched regulators in the EPC-CM are activated in a PI3K-dependent manner, and this could explain the impact of its inhibition in attenuating migration and invasion. However, VEGF inhibition with Bevacizumab and FGF inhibition using an anti-FGF2 antibody displayed only a modest effect on OS migration and MMP9 expression (inhibition of single factors in the EPC-CM). While the PI3K/AKT pathway is essential to promote growth and cell proliferation over differentiation in adult stem cells, it is overactive in many cancers [43]. A variety of factors have been found to enhance this central PI3K/AKT pathway, including angiopoietin (ANG2) and EGF [44]. These angiogenic factors might contribute, directly or indirectly, to the EPCs-mediated mechanism of OS development and progression that we describe in the current study. The approval of erdafitinib for the treatment of urothelial carcinoma with altered FGFR patterns in 2019 boosted FGF/FGFR cancer research [45]. However, these studies resulted in limited clinical benefits upon treatment with agents that solely block FGF/FGFR [46]. Previous in vitro studies showed that both inhibitors reduced PI3K/AKT levels. In a multiple myeloma cell culture, Bevacizumab attenuated VEGFR1 phosphorylation and reduced cell viability. This effect was found to be AKT-dependent [47]. In another study that aimed to investigate the effect of FGF-2 on neuronal apoptosis, it was found that a FGFR inhibitor (PD173074) abolished anti-apoptotic effects of rFGF-2 via suppression of AKT phosphorylation. In addition, anti-FGF-2 neutralizing antibody suppressed the expression of PI3K and p-AKT [48].

The effect of FGF on OS has not been extensively studied. In accordance with our findings (human and mouse samples), Xu and colleagues showed that 23 of 30 OS cases displayed a positive expression of FGF2 [41]. Together with attenuated migration due to FGF2 inhibition, we propose that FGF2 may constitute a druggable target candidate for OS therapy. An immunohistochemistry analysis of VEGF-A expression in OS specimens showed that it was associated with lung metastasis and diminished patient survival [49]. Moreover, VEGFR-2 was expressed in 65% of OS specimens and was associated with a pro-metastatic phenotype [49]. Our findings, as well as those of other studies [50,51,52,53], demonstrate a strong autocrine VEGF-based signaling which plays an essential role in providing self-sustained growth signals to OS cells. This may imply that anti-VEGF therapy acts in two modes of action: antiangiogenic therapy that inhibits tumor vasculature and an antineoplastic activity on the tumor cell population. The VEGF-A-directed antibody Bevacizumab is currently approved for use in combination chemotherapeutic regimens as the first and second line of treatment in colorectal cancer and as the first-line treatment in lung cancer [54]. Unfortunately, the benefits of anti-VEGF/VEGFR therapy observed in in vivo experiments were not translated into the clinic, and these drugs had a modest effect on human cancers [55]. For example, the combination of Bevacizumab and chemotherapy prolonged the survival of colon cancer patients for only 5 months [55].

As for OS, multi-target tyrosine kinase inhibitors (TKIs) have shown pre-clinical and clinical benefit in phase II clinical trials (cabozantinib, regorafenib, apatinib, sorafenib, and lenvatinib) [56]. In this respect, regorafenib and cabozantinib were used. Regorafenib is an oral TKI which inhibits VEGFR1/2/3, PDGFR, KIT, FGFR-1, and MET. In preclinical studies, regorafenib decreased OS cell growth and repressed VEGF-A and MMP9 gene expression [57]. Randomized clinical phase II studies showed increased progression-free survival (PFS) following regorafenib treatment in OS patients compared to OS patients that received a placebo. Cabozantinib targets VEGFR1/2/3, PDGFR, KIT, MET, RET, and AXL; it also demonstrated preclinical antitumor activity against OS. This TKI enhanced OS cell migration and proliferation via the ERK/AKT signaling pathways [58]. In clinical trials, cabozantinib increased the PFS to ~7 months [58].

Although a significant proportion of OS patients achieved temporary disease stabilization, mild responses were achieved with TKIs in monotherapy [58]. Consistent with these clinical results, our findings reveal that VEGF blockade had a negligible effect on OS migration and invasion. Therefore, a combined blockade of signaling pathways of several growth factors, including VEGF-A and FGF2, may be useful in inhibiting OS progression. These results are also in agreement with the National Comprehensive Cancer Network’s (NCCN) clinical guidelines that included multi-kinase inhibitors such as regorafenib and sorafenib as second-line therapy for OS [49].

## 5. Conclusions

In conclusion, our most remarkable finding is that EPCs promote in vitro OS invasion and migration in a paracrine manner via the PI3K/AKT signaling pathway. The EPCs secretome, which contains various factors, had a robust impact on OS migration and invasion when compared to the modest effect obtained with single factors such as VEGF-A and FGF2. Therefore, PI3K inhibition was more efficacious when compared to Bevacizumab. Due to the untoward toxicity and the wide range of possible side effects of PI3K inhibition, we believe that unraveling additional mediators in the EPC-CM is of paramount importance, particularly because of the potential of targeting these factors toward the development of novel targeted drugs for the future treatment of metastatic OS.

## Figures and Tables

**Figure 1 cancers-15-01818-f001:**
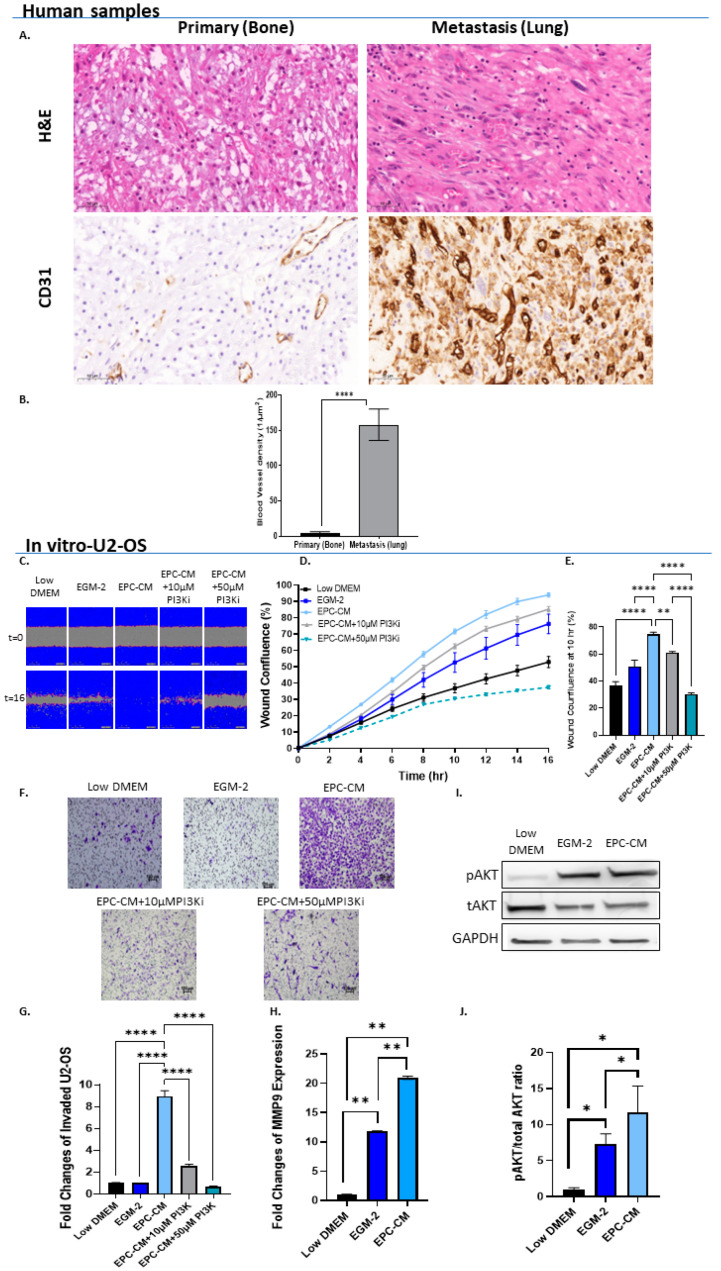
EPCs promote OS migration and invasion via PI3K/AKT signaling. (**A**) Representative H&E staining and CD31 immunolabeling of human OS lung metastasis (N = 18) and non-metastatic (N = 7) specimens. Cytoplasmic and membranous labeling patterns were obtained. Images were obtained under ×40 microscope magnification. Scale bar denotes 50 µm. (**B**) Quantitative analysis of blood vessel density (1/µm^2^) in metastatic vs. non-metastatic OS specimens. Vessel density was 30-fold higher in metastatic samples when compared to primary tumors; **** *p* < 0.0001. (**C**) Representative time-lapse microscopy images of U2-OS under different culture conditions: Low DMEM, EGM-2, EPC-CM, EPC-CM + 10 µM PI3Ki, and EPC-CM + 50 µM PI3Ki at 0 and 16 h. Images were obtained under ×10 magnification. Scale bar denotes 100 µm. (**D**) U2-OS migration rate. (**E**) Statistical analysis of U2-OS wound confluence under different culture conditions at 10 h. Wound confluence of cells cultured with EPC-CM was significantly higher compared to both controls (Low DMEM/MEM-Eagle and EGM-2); **** *p* < 0.0001. Addition of PI3Ki to EPC-CM significantly attenuated OS cell migration in a dose-dependent manner compared to EPC-CM alone; ** *p* < 0.01, and **** *p* < 0.0001. (**F**) Representative microscope images of invading U2-OS cells in Low DMEM/EGM-2/EPC-CM/EPC-CM + 10 µM PI3Ki and EPC-CM + 50 µM PI3Ki. (**G**) Quantitative analysis of invading U2-OS cells. U2-OS cell invasion was significantly higher in the EPC-CM group compared to EGM-2 and Low DMEM. PI3Ki significantly attenuated U2-OS cell invasion in a dose-dependent manner compared to EPC-CM; **** *p* < 0.0001. (**H**) Real-time PCR analysis. U2-OS cells were incubated with Low DMEM, EGM-2, or EPC-CM. Real-time PCR results revealed higher MMP9 expression level in the EPC-CM group compared to the controls; ** *p* < 0.01. (**I**) Western blot analysis images. U2-OS cells were incubated with Low DMEM, EGM-2, or EPC-CM. Phosphorylated AKT/total AKT and GAPDH antibodies were used to determine the pAKT/total AKT ratios. (**J**) Quantitative analysis of pAKT/tAKT ratios. EPC-CM significantly enhanced AKT phosphorylation in U2-OS cells compared to Low DMEM; * *p* < 0.05.

**Figure 2 cancers-15-01818-f002:**
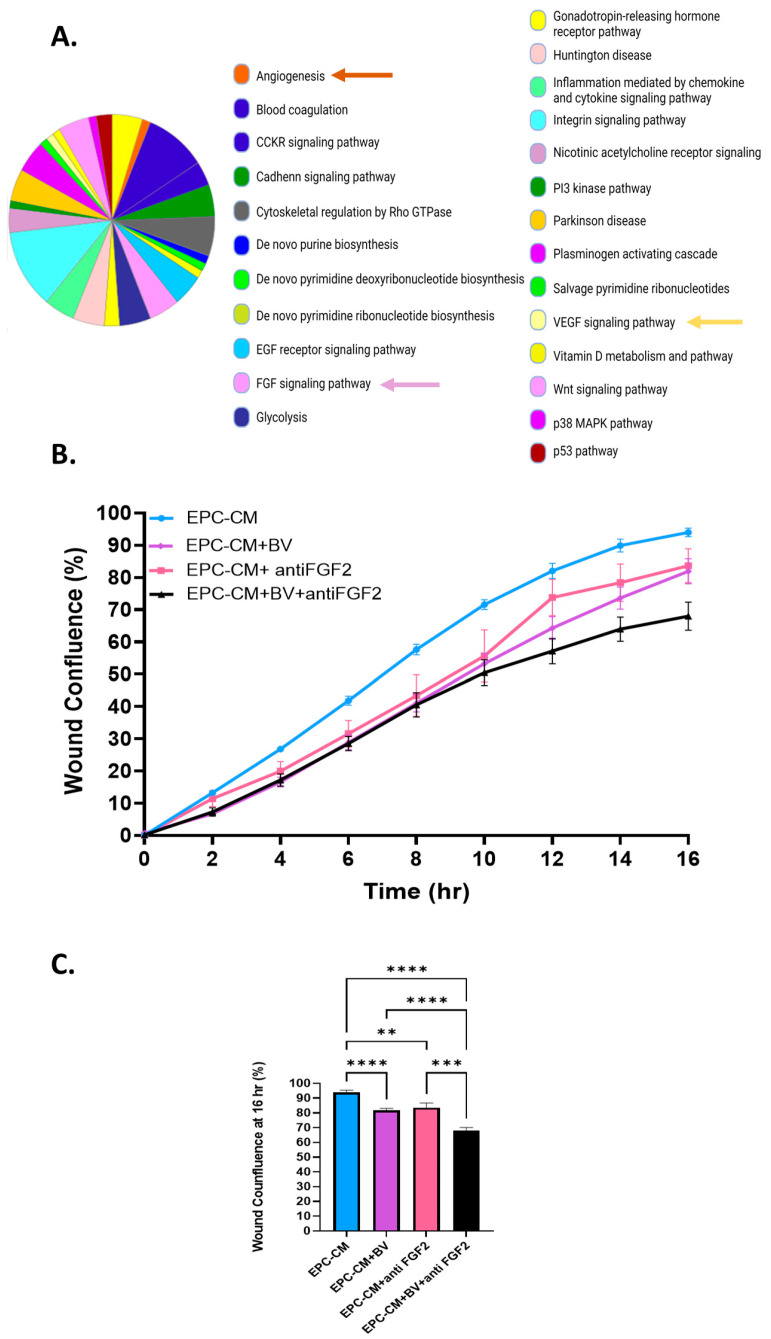
VEGF-A and FGF2 in the EPC-CM influence OS migration. (**A**) EPC-CM protein content. EPC-CM displays an angiogenic profile. CCKR, cholecystokinin receptors; EGF, epidermal growth factor; FGF, fibroblast growth factor; PI3 kinase, phosphoinositide 3 kinase; VEGF, vascular endothelial growth factor. (**B**) Migration rate of U2-OS cells cultured with EPC-CM, EPC-CM+ BV (2 mg/mL), EPC-CM+ anti FGF2 (0.08 µg/mL), and EPC-CM + BV (2 mg/mL) + anti FGF2 (0.08 µg/mL). (**C**) Statistical analysis of U2-OS wound confluence under different culture conditions. BV and anti FGF2 antibodies significantly decreased U2-OS cell migration compared to EPC-CM, while their combination reduced U2-OS migration even further; ** *p* < 0. 01, *** *p* < 0. 001, and **** *p* < 0.0001.

**Figure 3 cancers-15-01818-f003:**
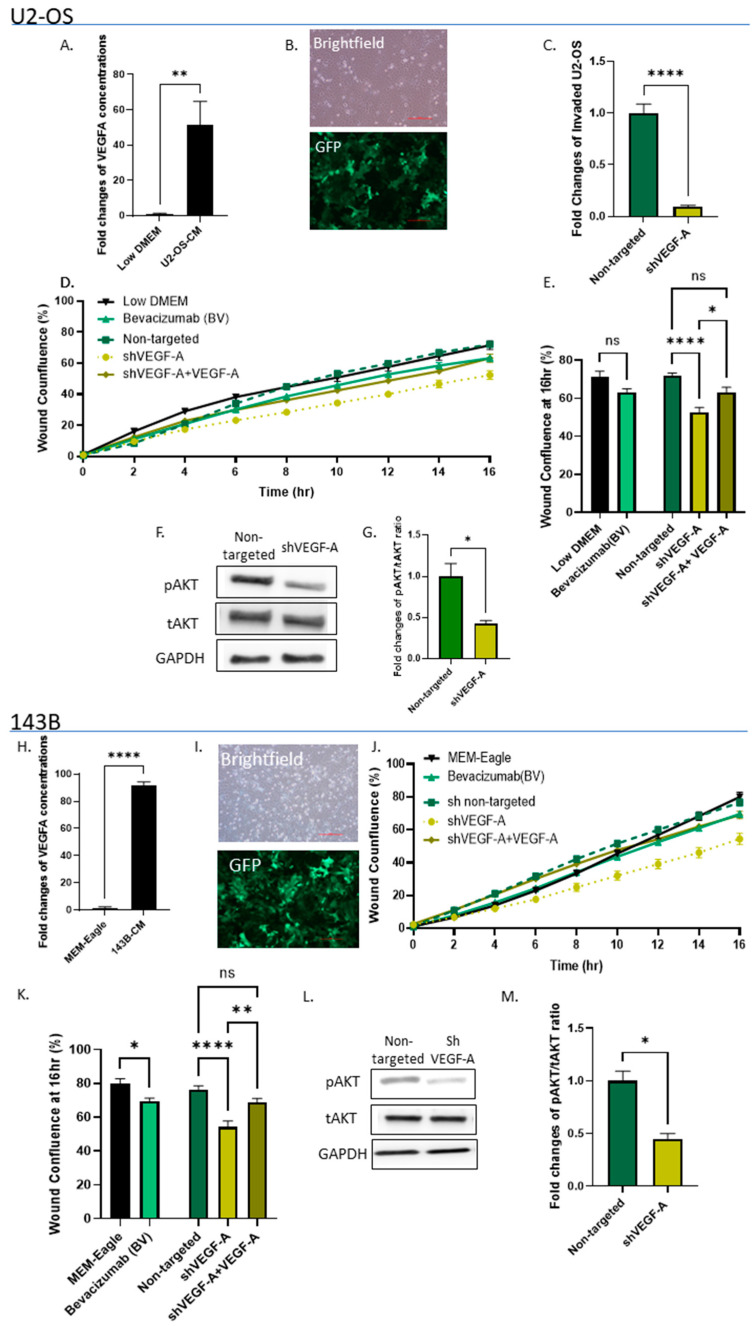
OS cells secrete VEGF-A, which promotes their invasiveness in an autocrine manner. (**A**,**H**) ELISA results reveal very high VEGF-A levels in U2-OS and 143B cell secretomes compared to standard growth media (Low DMEM/MEM-Eagle); ** *p* < 0.01, and **** *p* < 0.0001, respectively. (**B**,**I**) Bright field and fluorescence images of transduced U2-OS and 143B cells. Images were obtained under ×10 microscope magnification. Scale bar denotes 100 µm. (**C**) Quantitative analysis of invading shVEGF-A U2-OS cells vs. non-targeted control. ShVEGF-A U2-OS invasion was significantly lower compared to the non-targeted control; **** *p* < 0.0001. (**D**,**J**) Cell migration rates of shVEGF-A U2-OS and 143B cells and shVEGF-A U2-OS and 143B cells + VEGF-A (5 ng/mL) vs. non-targeted controls and U2-OS and 143B cells cultured with Low DMEM/MEM-Eagle + Bevacizumab (2 mg/mL) compared to standard medium alone (Low DMEM/MEM-Eagle). (**E**,**K**) Statistical analysis of wound confluence under different culture conditions at 16 h. Bevacizumab significantly attenuated U2-OS and 143B cell migration compared to Low DMEM/MEM-Eagle; * *p* < 0.05. ShVEGF-A inhibited U2-OS and 143B cell migration compared to non-targeted control (**** *p* < 0.0001); however, VEGF-A addition to shVEGF-A cells recovered this effect; ** *p* < 0.01, and * *p* < 0.05. (**F**,**L**) Western blot analysis of shVEGF-A U2-OS and 143B cells compared to non-targeted control. Phosphorylated AKT/total AKT and GAPDH antibodies were used to determine the pAKT/tAKT ratios. (**G**,**M**) Quantitative analysis of pAKT/tAKT ratios. VEGF-A silencing significantly decreased AKT phosphorylation in U2-OS and 143B cells compared to non-targeted cells; * *p* < 0.05.

**Figure 4 cancers-15-01818-f004:**
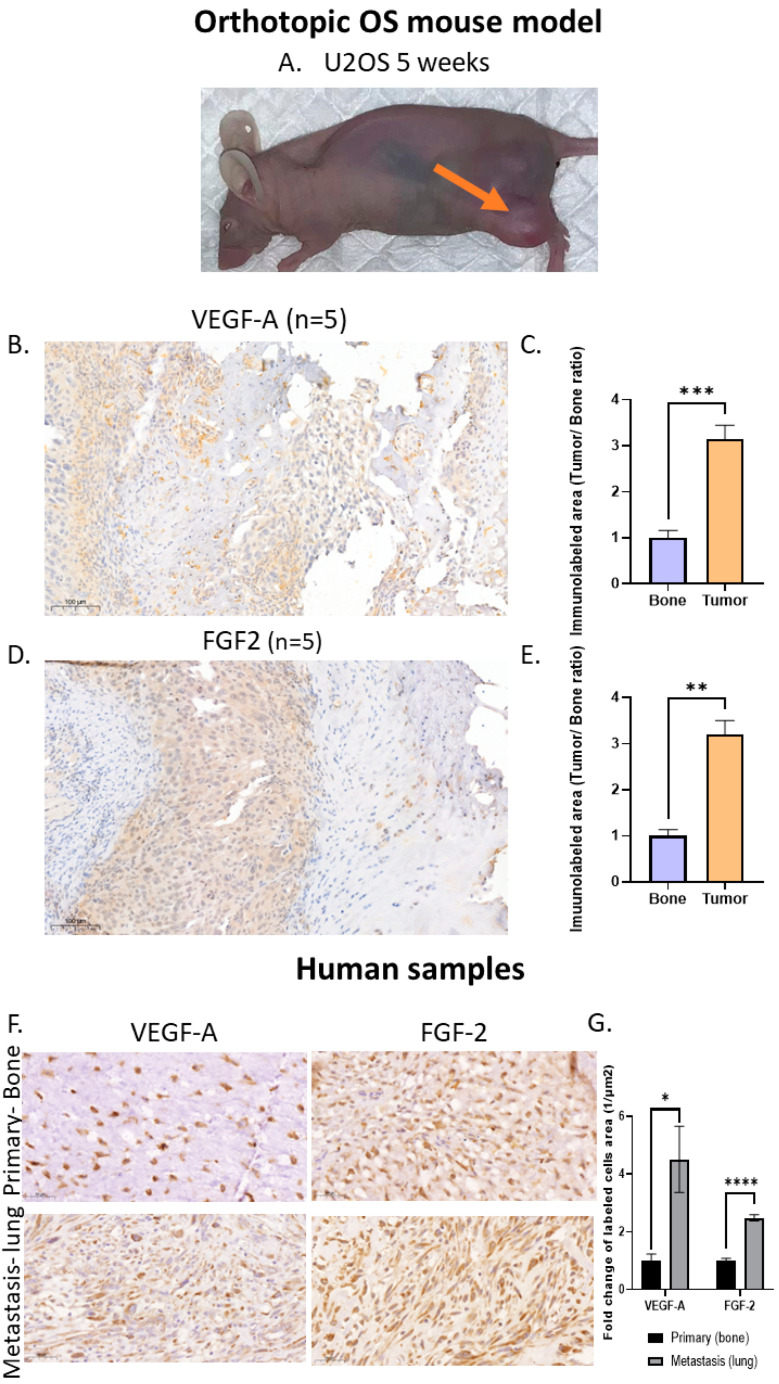
High VEGF-A and FGF2 labeling levels in an orthotopic OS mouse model and in human lung metastasis specimens: (**A**) U2-OS tumor image at 5 weeks and (**B**) anti-VEGF-A immunolabeling. Representative images of primary OS tumors in mice and nearby bone. The labeling pattern was mainly cytoplasmic; however, nuclear labeling was also noticed. Images were obtained under a ×40 magnification. Scale bar denotes 50 µm. (**C**) Quantitative analysis of immunolabeled area (tumor/bone ratio) of primary OS tumors in mice normalized to nearby bone. Primary OS tumors exhibit significantly higher VEGF-A levels compared to control bone; *** *p* < 0.001. (**D**) FGF2 immunolabeling. Representative images of primary OS tumors in mice and nearby bone. Images were obtained under ×40 magnification. Scale bar denotes 50 µm. (**E**) Quantitative analysis of immunolabeled area (tumor/bone ratio) of primary OS tumors in mice normalized to nearby bone. Primary OS tumors exhibit significantly higher FGF2 levels compared to control bone; ** *p* < 0.01. (**F**) Representative anti-VEGF-A and anti-FGF2 immunolabeling of human metastatic OS and non-metastatic OS patient specimens. Images showing nuclear labeling pattern. Microscope images were obtained under ×40 magnification. Scale bar denotes 50 µm. (**G**) Quantitative analysis of immunolabeled area (1/µm^2^) in metastatic vs. non-metastatic human OS specimens. Higher levels of VEGF-A (primary, N = 7; metastasis, N = 8) and FGF2 (primary, N = 6; metastasis, N = 14) were obtained in metastatic samples compared to primary tumors; * *p* < 0.05, and **** *p* < 0.0001.

**Figure 5 cancers-15-01818-f005:**
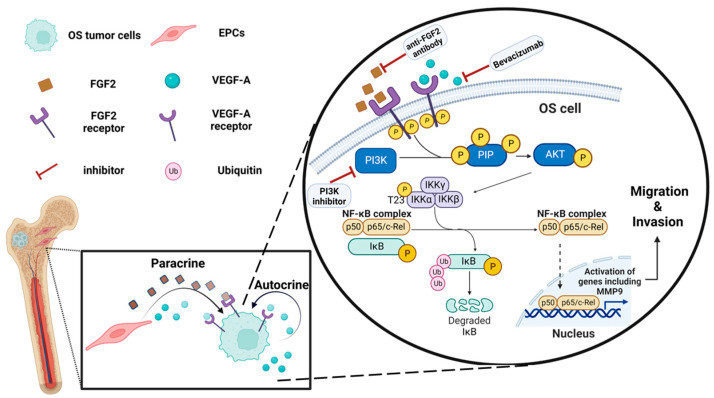
Proposed schematic illustration of the molecular mechanisms of the OS-EPC interaction: EPCs promote OS migration and invasion via the PI3K/AKT signaling pathway. EPCs secrete VEGF-A and FGF2 that activate, in a paracrine manner, PI3K/AKT signaling in OS cells, leading to upregulation of metastasis-related genes, including MMP9. OS cells secrete VEGF-A, which activates the pathway in an autocrine manner. PI3K inhibitor, FGF2 antibody, and Bevacizumab attenuate OS cell migration by inhibiting this pathway. The illustration was created with BioRender.com.

## Data Availability

The datasets used and/or analyzed during the current study are available from the corresponding author upon reasonable request.

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
