# Peer review of "Endothelial Progenitor Cells Promote Osteosarcoma Progression and Invasiveness via AKT/PI3K Signaling"

_cancers, 2023, doi:10.3390/cancers15061818_

Round 1

Reviewer 1 Report

In this manuscript, the authors investigated the role of stromal endothelial progenitor cells in the progression of osteosarcoma. The results of the study suggest that endothelial progenitor cells, in a paracrine manner, promote the progression and invasiveness of osteosarcoma, partly by activating PI3K/AKT though VEGF/FGF secretion. The study is well-designed, and the results are nicely presented. I recommend the article for publication given the comments below are addressed.

In the first part of the results, the authors used PI3K inhibitor, at 10 uM and 50 uM to investigate the effect of AKT/PI3K axis. How are the concentrations chosen and how strong is the inhibition of pAKT under these conditions? It would be helpful to report the inhibition of PI3K/pAKT, as well as MMP under these conditions.

AKT is downstream of PI3K and it would be more direct to use an AKT inhibitor to elucidate the control of invasiveness by AKT if an AKT inhibitor is also used, in addition to PI3K inhibitor. Have the authors tried to see the effects of an AKT inhibitor? Is it the same/different from PI3K inhibitor?

Using a positive control with a constitutively active PI3K or AKT could further help elucidate the role of PI3K/AKT.

In Figure 2, it would be helpful to also report the change in pAKT in control compared to bevacizumab and anti-FGF treated groups.

PI3K/AKT is also the downstream target of many other growth factors, including Angiopoietins. Could the authors briefly comment on other signaling pathways that could be mediating the activation of PI3K/AKT.

The authors have included a brief discussion of antiangiogenic treatment and VEGF targeting in cancer (lines 481 and on). In the context of the study, it would be helpful to summarize the preclinical and clinical development of VEGF/FGF targeting in the context of osteosarcoma.

Minor comment: are the error bars reported in the figures standard error of mean? It would be helpful to clarify. 

Reviewer 2 Report

Summary:

The aim of the paper is to evaluate the potential role of endothelial progenitor cells (EPCs) in osteosarcoma invasiveness and progression using in vitro assays as well as human patient and murine orthotopic tumor samples. This work nicely demonstrates enhanced in vitro invasion and migration of human osteosarcoma cells exposed to media EPC-conditioned media (EPC-CM). Proteomic analysis of the EPC-CM was used to identify potential EPC generated proteins and their related pathways that could be contributing to the behavioural changes seen in osteosarcoma cells exposed to the EPC-CM. The PI3K/AKT signalling pathway, VEGF-A, and FGF2 were implicated. The inclusion of inhibitory experiments with concurrent rescue was well done.

General Concepts:

For the most part, the in vitro assays were well designed. Additional methodological details and results for some of these in vitro assays will further strengthen the manuscript. Without additional details for some in vitro experiments, the conclusions will need to be altered (e.g. VEGF-A production by osteosarcoma cells). The weakest component of the paper is the immunohistochemistry (IHC). Controls are absent and there was inadequate description of the labeling. The current images with their annotations demonstrate that a pathologist was not consulted. As presented, the IHC results cannot be trusted.

The introduction is inadequate. The disease of interest is poorly defined with erroneous details and inappropriate references. MMP-9 and PI3K/AKT pathway were not introduced. Recent controversy regarding the origin of EPCs was not addressed.

Specific comments:

Immunolabeling and labeling - These are the appropriate terms when describing techniques that use antibodies to label antigens, not immunostaining and staining. This should be corrected throughout the document when describing any immunohistochemistry methods and results.

SIMPLE SUMMARY

Line 21: See comment below for Line 62. Also, does osteosarcoma really develop in the bone marrow per se?

ABSTRACT

Line 29-30: This statistic is not correct.

Line 30-31: See comment above for Line 21.

Line 46-48: This conclusion is overstated.

INTRODUCTION

Line 54-55: Osteosarcoma most commonly occurs in teenagers and young adults.

Line 56-58: This sentence requires additional words for clarity. What is the definition of long-term survival here? About 20% of patients have macrometastases at diagnosis; however, the majority of patients are suspected to have micrometastases. The citations for this sentence are not of adequate quality. Consider choosing alternate references and improving the clarity and accuracy of this sentence, as it is the only sentence referring to osteosarcoma prognosis.

Line 62: This fact has been question in the literature in recent years (1, 2). The cited reference for this fact is from 2007. Please discuss the controversy or make a less definitive statement.

1. Fujisawa T et al. Endothelial progenitor cells do not originate from the bone marrow. Circulation 2019;140(18):1524–1526.

2. Salybekov AA et al. Characterization of endothelial progenitor cell: past, present, and future. Int. J. Mol. Sci. 2022;23(14):7697. 

Line 63-65: Are these the most current positive and negative markers for EPCs? The paper cited is from 2007.

Line 97-100: How much EGM-2MV was added to the EGM-2?

Lines 119-122: Were there no supplements added to the standard growth medium for U2OS and 143B? This seems odd, especially since additives were supplemented into the media used for the invasion assay (Line 134-135).

MATERIALS AND METHODS

Line 92: At which facility did this occur? How was consent acquired?

Line 146: The standard culture media for 143B was previously defined as “MEME”; however, “MEM-Eagle” is used from this point forward.

Line 147: Wouldn’t it be cell migration into the wound, not toward the wound? Minor point for consideration.

Line 149: Which cell lines were used and for how long under the provided conditions?

Line 155: Which cell lines? Under which conditions?

Line 165: “EPC-CM/EGM-2” should these be separated by a comma instead of a “/”.

Line 161-162: Is this an appropriate annotation for supplementary data?

Line 185: This section lacks sufficient detail to understand how these experiments were done. Which cell lines, under which conditions, and for what amount of time were used?

Line 186: All of the supernatant? Or a specific volume?

Line 201-202: Which type of media?

Line 202-204: 2 x sentences duplicated, delete.

Line 245: Were the diagnoses of the human samples confirmed by a pathologist? Which samples did you have from the patients with metastases: the primary, the met, or both? Or did it vary between patients?

Line 249 (Immunohistochemistry): The demineralization status of the tissues is not provided. Tissue demineralization can alter IHC results and must be considered. Negative and positive controls are not mentioned. At a minimum, a description of the negative controls is required. There is no indication of who evaluated the IHC; was a pathologist consulted? Why do the number of cases included for the metastatic cases vary by antibody?

Line 229-231: Please clarify the use of the 27G vs 29G syringe during orthotopic implantation.

Line 236: “tape meter”, confirm accurate use of terminology.

Line 236-238: How were isolated limbs weighed in live unanesthetized animals? Was this data solely used for body weight tracking?

Line 241-242: What is the significance of 5 vs 7 weeks post-implantation? Were these groups ever compared?

Line 265-266: Consider omitting “Hematoxylin staining was used for general morphology” as it is unnecessary.

RESULTS

Line 279: No results for EPC isolation, expansion, or characterization are provided.

Line 280: Add “in vitro” after “OS” or after “invasion”.

Line 285-286: Uncertain of the validity of these results (see specific comments for Fig. 1)

Line 292-302: Should all sentences referring to result not have figure references?

Line 314 (Figure 1):

1A – These images are not of adequate quality. The tissue morphology cannot be determined. The H&E image is not matched to the IHC image. Based on the images provided, I do not have confidence that the IHC was interpreted properly.

1E – Why is this analysis done at 10 hours when 1C and 1D are for 16 hours?

1F – These images are of inadequate size and quality making them more difficult to interpret. Consider making them larger.

Line 341-342: Pathways with greater contributions to the protein content were identified also.

Line 342-343: Where is this data? Why is it not provided, even as supplemental material?

Line 343-346: This is not in the methods, nor are the results provided for any proteins other than VEGF-A or FGF2.

Line 350-351: Consider referencing figure and including statistical significance.

Line 351-352: Consider including statistical significance.

Line 356 (Figure 2):

2A – Consider making the pie chart an actual circle, it looks distorted.

Line 357-358: Only the angiogenic profile is mentioned here, but the FGF and VEGF signaling pathways are also highlighted in the figure. Also, consider providing definitions for the acronyms in the figure.

Line 362: “***” is not defined.

Line 364-368: The Materials and Methods were insufficient for this experiment; therefore, it is difficult to tell what comparison is being made here. I believe there might be a comparison missing here. The level of VEGF-A in the conditioned media prior to incubation with the osteosarcoma cell lines should be provided. However, without the experimental details being provided I cannot be sure what comparison is being made here (Fig 3A & 3H). Based on what has been provided, this result, “both OS cell lines were found to secrete very high levels of VEGF-A”, has not been demonstrated.

Line 390: The figure legend indicates 10 hours whereas the figure indicates 16 hours for Figure 3E and 3K.

Line 398-410: IHC labeling patterns not described. What part of the cell is positive? Which cell types are positive?

Line 401-402: Did all mice develop tumors? What was the average tumor size at endpoint? According to the methods, 11 mice were implanted and sacrificed at 2 different time points, but the results mention 5 mice total. Please clarify.

Line 403-406: Based on the image provided, the validity of the IHC results are questionable. (see comments for Figure 4).

Line 406-410: Inadequate detail provided. A table with IHC results and cases included for each antibody is highly recommended. Is it appropriate to compare primary and metastatic tumors in unmatched samples? If you are interested in invasion and migration, should you not have compared primary tumors from patients with and without metastases? Can you justify your choice of samples for comparison?

Line 411 (Figure 4):

4A – The implantations were described as being in the distal femur; however, the arrows are pointing to the proximal femur or pelvis. There is a tumor in the distal femur of the 7-week mouse, but the arrow is not identifying the tumor. I see no obvious tumor in the 5-week mouse. Have the arrows been inadvertently shifted to the right?

4B – This image is not of adequate quality, nor does it appear to show evidence of metastasis. Although the quality is low, I believe it represents an off-centre portion of a bronchiole, not a tumor. The arrow appears to be pointing at airway epithelium. (*consider using IHC to help identify metastases – e.g. human mitochondrial antibody).

4C – This is one of the better-quality histology/IHC pictures in this manuscript; however, it is still slightly out of focus. Unfortunately, the image appears to be almost entirely tumor; I suspect that the bone present is neoplastic bone, although it could be remnant bone that has been invaded by the tumour. The authors have not described the labeling pattern. I see limited positive immunolabeling in this image.

4E – This image does not appear to contain any tumor. The “T” is labeling blood and hematopoietic cells. This section appears to have been taken through a growth plate. There is no description of labeling pattern in the figure legend.

4F – “tatio” must be corrected.

4G – The images in the panel are of inadequate quality. The architectural nor cellular details can be evaluated. No description of labeling pattern is described in the figure legend.

DISCUSSION

Line 429-431: Consider a more recent reference than a publication from 2010.

Line 434-435: The results as they stand are in vitro only. Therefore, this is an overstatement.

Line 438-439: Does a 2012 publication count as a “recent report”?

Line 456-458: Expand for clarity.

Line 472-474: More details required to establish VEGF production by osteosarcoma cells in this study.

CONCLUSIONS

Line 490-491: This is an overstatement; you should add the term “in vitro” to this sentence.

Line 499 (Figure 5): More details are required to establish the level of VEGF-A production but osteosarcoma cells.

Line 500: I think you need to say “proposed” in here as not all of the components of this pathway were addressed in this study.

Line 500-505: In this paper, none of the components of the pathway between pAKT and MMP-9 are examined.

Reviewer 3 Report

Ofri Doppelt Flikshtain and co-authors investigated the role of EPCs in osteosarcoma progression. The topic might be interesting, but the novelty is low. 

- improve the quality of all figures. 

- the authors should explain the failure of EPC-CM on tumor cells proliferation induction. EPC-CM is enriched of growth factors involved in cells mitosis. 

- Fig. 1C. How did authors inhibit OS proliferation during wound healing assay?

- Fig. 1H. MMP (2 and 9) activity should be analized by performing zymography, as well as the protein expression of TIMP.

- Fig. 1 I. Have the authors checked the activation of MAPK (ERK1/2) signaling upon same condition? This pathway plays crucial role in tumor progression and invasion.

- The dosage of VEGF (pg/mL) by ELISA in EPC-CM should be performed. 

- EGM-2 in very enriched medium. It may contribute to OS cells migration

- I don’t understand the purpose of xenograft experiment. The authors analyzed the VEGF-A and FGFb positivity. Which is the contribution of EPC in this experiment? It is expected a greater expression of VEGF-A and FGFb in tumor compared with nearby health bone tissue. Which is the novelty?

Round 2

Reviewer 2 Report

General comments:

Overall, the authors did a nice job addressing my concerns in the previous review. There are a few additional comments regarding the in vitro experiments, but they are all minor.

The largest remaining issue is the immunohistochemistry (IHC). The authors provided additional images of a much higher quality in this version of the manuscript, which was greatly appreciated. However, the IHC was not properly evaluated. As they stand, none of the IHC results should be published. The quality of the IHC is not the issue, it looks very well done.

For the CD31 IHC, the issue is that the bone samples were demineralized and the lung samples were not. A control experiment must be completed to establish that demineralization does not alter the labeling affinity. Without this, you cannot compare the labeling in tissues that have been demineralized and those that have not been demineralized.

The issue with the interpretation of the VEGF-A and FGF2 IHC is that it appears as though the positively labeled tissue area is being evaluated and not the positively labeled cells. This is not how the IHC should be evaluated in this case, as the amount of matrix will ultimately predetermine the results.

All of the IHC must be re-evaluated, none of the results have any biological relevance as they stand. All conclusions reached from the IHC data must be removed from the paper if the analysis is not properly repeated.

Specific comments:

Line 27: Should this be “deciphering a potential underlying molecular mechanism”?

Line 40: “orthotopic”

Line 51-53: Check sentence, difficult to interpret the edits that have been made.

Line 60-61: The end of the sentence about the “second peak” did not have to be deleted.

Line 136: Name of supplementary file doesn’t match. CXCR4 results are reported in the supplementary material but this marker is not mentioned in the introduction or materials and methods. Some EPC cell lines were positive for CD14 and CD45, but this is never discussed. How did the authors decide these were EPCs?

Line 189: This is written in section 2.6 of the methods section, so this line shouldn’t refer to section 2.6 of the methods section.

Line 285-289: I am still curious why there is a 5 week and a 7 week timepoint, and this is not discussed.

Line 289: Were they “collected randomly” from the 12 mice total? A mix of the 5 and 7 week timepoint mice?

Line 291-292: Please clarify that the n=18 are the metastases and not the primary tumors from the patients with metastases. For example: “Histological samples from the primary tumors (N=7) of non-metastatic OS patients and the metastatic tumors (N=18) from OS patients with lung metastases.”

Line 296: “Stained” is the appropriate term for H&E. Revert to original.

Line 298: No antigen retrieval information has been provided for any of the antibodies.

Line 309: What is N=65 referring to?

Line 315: “Stained” is the appropriate term for H&E. Revert to original.

Line 318: Negative control image only provided for VEGF-A and not for CD31 or FGF2.

Line 335-337: Do you know that the patients for the primary tumors never metastasized? You should clarify here that you are comparing primary tumors to metastatic tumors (as opposed to the primary tumors from patients who did vs did not develop metastases).

Line 370 (Figure 1): The appearance of the images in this figure is greatly improved. Font sizes in figure labels for A are not consistent. Were the lung metastases also subjected to demineralization? If not, this could explain your labeling differences. This must be addressed for these results to be valid. You must demonstrate that the demineralization has no effect on CD31 immunolabeling.

Line 372: “Stained” is the appropriate term for H&E. Revert to original.

Line 373: Consider substituting “membrane” for “membranous”.

Line 426: Thank you for adding additional details to the materials and methods. Initially, I thought you used EPC-CM on the OS cells and then measured the VEGF-A.

Line 442: Why was neither VGFA or FGF2 or the combination added to non-conditioned media to test the effects on migration or invasion? But only on MM9 expression?

Lines 461-500: Please refer to my general summary comments. This IHC was not evaluated properly.

Line 476-479: Was 143B an intravenous study or an orthotopic study? This 143B in vivo experiment was not described in the materials and methods. Why are lung metastases from the U2OS mice not demonstrated? In the previous version, it was stated that 100% of mice had metastases. Why was IHC no performed on the metastatic tumors from the mice?

Line 481 (Figure 4): The image quality is much better. However, the IHC images (B, D, F) do not appear to accurately represent the quantitative data provided (C, E, G). It appears as though the bone matrix is being included as negative. However, only the cells should be evaluated. For example, in 4F-Primary-Bone almost 100% of cells are positive, but the authors are indicating in 4G that primary bone tumors had significantly less positive labeling than lung metastases. Either the incorrect images were selected or the immunohistochemistry has not been properly evaluated. Is the comparison to normal necessary? Is comparing primary to metastatic OS important to this work? Migration and invasion is more useful to neoplastic cells in the primary tumor than the tumor cells that have already metastasized to the lung.

Line 494: “staining” does not have to be omitted here, it is accurate use of the term for H&E.

Line 523: Replace “In this study” with “In the current study”.

Line 565: Where is this experiment described? I see the MM9 expression in Figure 4S, but where is the migration assay data for single agent VEGF-A and FGF2 treatment?

Reviewer 3 Report

The authors have well addressed my comments

Author Response

We thank the reviewer for his valuable comments. 

Round 3

Reviewer 2 Report

General comments:

Overall, the authors adequately addressed most of the minor comments from my previous review; however, the major issue was not addressed. The remainder of the general comments are essentially unchanged from my previous review.

The largest remaining issue is the immunohistochemistry (IHC). The IHC was not properly evaluated. As they stand, none of the IHC results should be published. The quality of the IHC is not the issue, it looks very well done.

For the CD31 IHC, the issue is that the bone samples were demineralized and the lung samples were not. A control experiment must be completed to establish that demineralization does not alter the labeling affinity. Without this, you cannot compare the labeling in tissues that have been demineralized and those that have not been demineralized.

The issue with the interpretation of the VEGF-A and FGF2 IHC is that it appears as though the positively labeled tissue area is being evaluated and not the positively labeled cells. This is not how the IHC should be evaluated in this case, as the amount of matrix will ultimately predetermine the results.

All of the IHC must be re-evaluated, none of the results have any biological relevance as they stand. All conclusions reached from the IHC data must be removed from the paper if the analysis is not properly repeated.

Specific comments:

***Line numbers are from the most up to date clean version of the manuscript.

Line 98: Comment repeated from previous review - CXCR4 results are reported in the supplementary material but this marker is not mentioned in the introduction or materials and methods.  Some EPC cell lines were positive for CD14 and CD45, but this is never discussed. How did the authors decide these were EPCs?

Line 278-280: You have not addressed the fact that demineralization can alter immunolabeling. Because you are comparing IHC in non-demineralized tissue to IHC in demineralized tissue, this is necessary. Please see my comments from the last review.

Line 302-303: The is a Fig 1S-A-B, but I can’t see Fig 1S-1-2 (maybe I do not have the updated version). Unless the same antigen retrieval was used for all antibodies, then you need a separate negative control for each.

Line 448-450: Instead of explaining the 5- vs 7-week timepoint, the 7-week timepoint has been completely omitted. This does not seem appropriate, as previously 5 of 12 mice were chosen for immunohistochemistry, now 5 of 6 mice were chosen. It makes no sense. This seems inappropriate if IHC was performed on tissues from some of the mice from the 7-week timepoint which has now disappeared.

Line 450-457: Please see my comments from the previous review. The IHC issues have not been addressed and the IHC has not been properly evaluated. Yes, the IHC shows that the cells express FGF2 and VEGF-A; however, the comparisons do not appear to be accurate based on the images provided.
